# Long-term cost-effectiveness of interventions for obesity: A mendelian randomisation study

**Sean Harrison**[ID][1,2]*, **Padraig Dixon**[ID][1,2], **Hayley E. Jones**[ID][2], **Alisha R. Davies**[3], **Laura D. Howe**[ID][1,2☯], **Neil M. Davies**[ID][1,2,4☯]

1 MRC Integrative Epidemiology Unit (IEU), Population Health Sciences, Bristol Medical School, University of Bristol, Bristol, United Kingdom, 2 Population Health Sciences, Bristol Medical School, University of Bristol, Bristol, United Kingdom, 3 Research and Evaluation Division, Public Health Wales NHS Trust, Cardiff, United Kingdom, 4 K.G. Jebsen Center for Genetic Epidemiology, Department of Public Health and Nursing, NTNU, Norwegian University of Science and Technology, Trondheim, Norway

☯ These authors contributed equally to this work.
* sean.harrison@bristol.ac.uk

## Abstract

### Background

The prevalence of obesity has increased in the United Kingdom, and reliably measuring the impact on quality of life and the total healthcare cost from obesity is key to informing the cost-effectiveness of interventions that target obesity, and determining healthcare funding. Current methods for estimating cost-effectiveness of interventions for obesity may be subject to confounding and reverse causation. The aim of this study is to apply a new approach using mendelian randomisation for estimating the cost-effectiveness of interventions that target body mass index (BMI), which may be less affected by confounding and reverse causation than previous approaches.

### Methods and findings

We estimated health-related quality-adjusted life years (QALYs) and both primary and secondary healthcare costs for 310,913 men and women of white British ancestry aged between 39 and 72 years in UK Biobank between recruitment (2006 to 2010) and 31 March 2017. We then estimated the causal effect of differences in BMI on QALYs and total healthcare costs using mendelian randomisation. For this, we used instrumental variable regression with a polygenic risk score (PRS) for BMI, derived using a genome-wide association study (GWAS) of BMI, with age, sex, recruitment centre, and 40 genetic principal components as covariables to estimate the effect of a unit increase in BMI on QALYs and total healthcare costs. Finally, we used simulations to estimate the likely effect on BMI of policy relevant interventions for BMI, then used the mendelian randomisation estimates to estimate the cost-effectiveness of these interventions.

A unit increase in BMI decreased QALYs by 0.65% of a QALY (95% confidence interval [CI]: 0.49% to 0.81%) per year and increased annual total healthcare costs by £42.23 (95% CI: £32.95 to £51.51) per person. When considering only health conditions usually considered in previous cost-effectiveness modelling studies (cancer, cardiovascular disease,

**Data Availability Statement:** The empirical dataset is archived with UK Biobank and available to individuals who obtain the necessary permissions from the study's data access committees, with data

accessible from https://www.ukbiobank.ac.uk/. The code used to clean and analyse the data is available here: https://github.com/sean-harrison-bristol/Robust-causal-inference-for-long-term-policy-decisions.

**Funding:** The Medical Research Council (MRC) and the University of Bristol support the MRC Integrative Epidemiology Unit [MC_UU_00011/1]. NMD is supported by an Economics and Social Research Council (ESRC) Future Research Leaders grant [ES/N000757/1] and the Norwegian Research Council Grant number 295989. LDH is supported by a Career Development Award from the UK Medical Research Council (MR/M020894/1). PD acknowledges support from a Medical Research Council Skills Development Fellowship (MR/P014259/1). This work is part of a project entitled 'social and economic consequences of health: causal inference methods and longitudinal, intergenerational data', which is part of the Health Foundation's Social and Economic Value of Health Programme (Grant ID: 807293). The Health Foundation is an independent charity committed to bringing about better health and health care for people in the UK. The funders had no role in study design, data collection and analysis, decision to publish, or preparation of the manuscript. This publication is the work of the authors, who serve as the guarantors for the contents of this paper.

**Competing interests:** The authors declare they have no conflicts of interest.

**Abbreviations:** BMI, body mass index; CI, confidence interval; GWAS, genome-wide association study; HES, hospital episode statistics; HFSS, high fat, sugar, and salt; IQR, interquartile range; IVW, inverse-variance weighted; PRS, polygenic risk score; QALY, quality-adjusted life year; RCT, randomised controlled trial; SD, standard deviation; SNP, single nucleotide polymorphism.

cerebrovascular disease, and type 2 diabetes), we estimated that a unit increase in BMI decreased QALYs by only 0.16% of a QALY (95% CI: 0.10% to 0.22%) per year.

We estimated that both laparoscopic bariatric surgery among individuals with BMI greater than 35 kg/m$^2$, and restricting volume promotions for high fat, salt, and sugar products, would increase QALYs and decrease total healthcare costs, with net monetary benefits (at £20,000 per QALY) of £13,936 (95% CI: £8,112 to £20,658) per person over 20 years, and £546 million (95% CI: £435 million to £671 million) in total per year, respectively.

The main limitations of this approach are that mendelian randomisation relies on assumptions that cannot be proven, including the absence of directional pleiotropy, and that genotypes are independent of confounders.

## Conclusions

Mendelian randomisation can be used to estimate the impact of interventions on quality of life and healthcare costs. We observed that the effect of increasing BMI on health-related quality of life is much larger when accounting for 240 chronic health conditions, compared with only a limited selection. This means that previous cost-effectiveness studies have likely underestimated the effect of BMI on quality of life and, therefore, the potential cost-effectiveness of interventions to reduce BMI.

## Author summary

### Why was this study done?

- The prevalence of obesity has increased in the United Kingdom, and reliably measuring the impact on quality of life and the total healthcare cost from obesity is key to informing the cost-effectiveness of interventions that target obesity, and determining how much additional healthcare funding may be required should the trend of increasing obesity continue.

- Current methods of examining cost-effectiveness of interventions for obesity may be subject to confounding and reverse causation, and previous studies also typically only use a limited number of health conditions to estimate the effects of BMI on quality of life, potentially underestimating the effects of BMI.

- The aim of this study is to elucidate a new approach using mendelian randomisation for estimating the cost-effectiveness of interventions that target body mass index (BMI), which may be less affected by confounding and reverse causation than previous approaches.

### What did the researchers do and find?

- Using mendelian randomisation, we estimated that a unit increase in BMI decreased quality-adjusted life years (QALYs) by 0.65% of a QALY per year and increased annual total healthcare costs by £42.23 per person.

- Using these results and simulations, we estimated that, compared to no intervention and over 20 years, people aged 40 to 69 years in England or Wales with a BMI over 35 kg/m$^2$ receiving laparoscopic bariatric surgery would have, on average, an increase of 0.92 QALYs and a decrease in total healthcare costs of £5,096 per person.

- We also estimated that restricting volume promotions for high fat, salt and sugar products would, across the 21.7 million adults aged 40 to 69 years in England and Wales, increase QALYs by 20,551 per year and decrease total healthcare costs by £137 million per year, and that between 1993 and 2017 in England and Wales, the increase in BMI of people aged 40 to 69 years led to a decrease of 1.13% of a QALY per year and an increase in annual healthcare costs of £69 per person.

## What do these findings mean?

- Mendelian randomisation can be used to estimate the impact of interventions on quality of life and healthcare costs and is likely less biased than existing observational methods.

- Interventions for BMI are likely to be cost-effective, possibly more so than previously anticipated using simulation methods that restrict the effect of changes in BMI on health conditions to cancer, cardiovascular disease, cerebrovascular disease, and type 2 diabetes.

## Introduction

Between 1993 and 2017 in England, the prevalence of obesity in adults aged 40 to 69 years, defined as a body mass index (BMI) of over 30 kg/m$^2$, rose from 13% to 27% in men and 16% to 30% in women, as estimated by the Health Survey for England [1,2]. Obesity is associated with many chronic illnesses, such as hypertension, coronary artery disease, type 2 diabetes, dyslipidaemia, metabolic liver disease, renal and urological diseases, sleep apnoea, osteoarthritis, psychiatric comorbidity, gastro-oesophageal reflux disease, and various cancers [3–7]. Reliably measuring the impact on quality of life and the total healthcare cost from obesity is key to informing the cost-effectiveness of interventions that target obesity, and determining how much additional healthcare funding may be required should the trend of increasing obesity continue. For example, prominent recent policy interventions such as the introduction in England of a tax on sugar-sweetened beverages were motivated in part by a desire to avoid some of the long-term consequences of obesity on individuals and the healthcare system [8].

Previous studies examining the cost-effectiveness of interventions for obesity tended to fall into 3 broad categories: (a) randomised controlled trials (RCTs), typically with relatively short-term durations of follow-up [9]; (b) cohorts, typically retrospective [10–13]; and (c) decision analytic and related simulation models [10,12,14–18]. These studies estimated the impact on quality-adjusted life years (QALYs) and the total healthcare cost of different interventions, such as bariatric surgery, and thus estimated whether the intervention was likely to be cost-effective. **Fig 1A–1C** show schematic representations of each type of study, **Table 1** summarises their strengths and limitations, and **S1 Text** gives more information about each type of study.

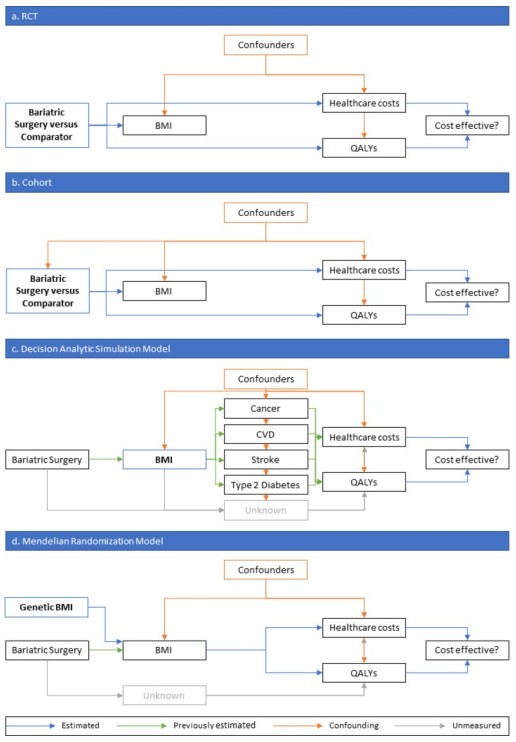

**Fig 1. Schematic representation of different methods of estimating cost-effectiveness of bariatric surgery.** The intervention or exposure for each analysis is in the blue box with bold text. Blue arrows represent what is estimated in each study, while green arrows represent estimates from previous studies used to inform the study. (**a**) The estimate of cost-effectiveness is not confounded as the intervention is randomised. (**b**) The estimate of cost-effectiveness could be confounded as receiving bariatric surgery is not randomly assigned. (**c**) The estimate of cost-effectiveness could be confounded, as could be the estimates from previous studies, there may be effects of bariatric surgery on QALYs and healthcare costs that do not go through BMI, and there may be effects of BMI on QALYs and healthcare costs that do not go through the modelled health conditions. (**d**) the estimate of cost-effectiveness is less likely to be affected by confounding, as genetic variants are randomly distributed within families at conception, though there may be effects of bariatric surgery on QALYs and healthcare costs that do not go through BMI. BMI, body mass index; CVD, cardiovascular disease; QALY, quality-adjusted life year; RCT, randomised controlled trial.

Briefly, RCTs with economic evaluations provide causal evidence for cost-effectiveness but are expensive and time consuming to perform, while cohort studies are observational and decision analytic simulation models rely on observational evidence that may be subject to confounding and reverse causation that may bias estimates of cost-effectiveness. Decision analytic simulation models also routinely include only a limited selection of health conditions that BMI may affect, meaning the true costs of obesity may be underestimated.

The aim of this study is to elucidate a new approach using mendelian randomisation [19,20] for estimating the cost-effectiveness of interventions that target BMI (**Fig 1D**). This approach uses observational data, but by using genetic information as an instrumental variable, the risk of bias through confounding and reverse causation is reduced compared with other methods using observational data [21–23]. This can give more causal estimates of cost-effectiveness, approximating an RCT of different BMI levels assigned at birth, but with the advantage of estimating at low cost the long-term causal effects of an intervention, rather than shorter-term effects measured during a (usually) limited period of follow-up measured in an economic evaluation conducted alongside an RCT.

In this paper, we estimate the causal effect of a unit increase in BMI on both QALYs and total healthcare costs in UK Biobank [24] using mendelian randomisation. We then

**Table 1. The strengths and limitations of different methods to estimate the cost-effectiveness of interventions.**

| Methods | Strengths | Limitations |
|---|---|---|
| RCT, with economic evaluation | • Causal effect estimates | • Expensive<br>• Time consuming<br>• Often limited follow-up<br>• Study sample may not be representative of target population |
| Cohort | • Follow-up may be long<br>• Potentially less expensive than RCTs<br>• A single study can test multiple hypotheses | • Estimates may be biased by confounding and reverse causation (control group not "exchangeable" with intervention group) |
| Decision analytic simulation models | • Inexpensive<br>• Follow-up not limited<br>• Flexible | • Estimates may be biased by confounding and reverse causation<br>• Using only limited health conditions (cancer, cardiovascular disease, cerebrovascular disease, and type 2 diabetes) as mediators of the effect of the exposure on the outcome may cause bias<br>• Effect estimates are for a change in the exposure, not an intervention for the exposure, and therefore are best applied to an intervention that affects the exposure across the life course |
| Mendelian randomisation | • Follow-up may be long<br>• Potentially less expensive than RCTs<br>• A single study can test multiple hypotheses<br>• Estimates less liable to confounding and reverse causation than cohort and decision analytic simulation studies | • Low statistical power; requires very large sample sizes<br>• Effect estimates are for a change in the exposure, not an intervention for the exposure, and therefore are most relevant to proxy an intervention that affects the exposure across the life course |

RCT, randomised controlled trial.

demonstrate how the results from this approach can be used to estimate the cost-effectiveness of prominent and widely used interventions aimed at reducing BMI (with bariatric surgery and restricting volume promotions for high fat, sugar, and salt (HFSS) products as case studies), estimate the increased healthcare cost of the rise in BMI in England and Wales between 1993 and 2017, and estimate the total cost of the BMI profile of England and Wales in 2017 versus a hypothetical profile where no one has a BMI above 25 kg/m$^2$.

## Methods

We used mendelian randomisation to estimate the causal effect of BMI on QALYs and total healthcare costs per year. For a guide to mendelian randomisation for clinicians, please see Davies and colleagues [20], and for a lay description, please see Harrison and colleagues [25]. Briefly, we generated a polygenic risk score (PRS) for BMI (a weighted score of genetic risk for higher BMI using common genetic variants), which we used as a proxy for BMI in the mendelian randomisation analyses.

### Population

UK Biobank is a population-based health research resource consisting of approximately 500,000 people, who were recruited between the years 2006 and 2010 from 22 centres across the United Kingdom [24]. Medical data from hospital episode statistics (HES) has been linked to all participants up to 31 March 2017, and primary care (general practice) data have been linked to UK Biobank participants registered with GP surgeries using EMIS Health (EMIS Web) and TPP (SystmOne) software systems, also up to 31 March 2017. The study design, participants, and quality control methods have been described in detail previously [26–28]. UK Biobank received ethics approval from the Research Ethics Committee (REC reference for UK Biobank is 11/NW/0382). Genotyping information is available in **S2 Text**, with further information available online [29].

We restricted the main analyses to unrelated individuals of white British ancestry living in England or Wales at recruitment, with a measured BMI value. Full details of inclusion criteria and genotyping are in **S2 Text**. After exclusions, 310,913 participants remained in the main dataset. Of these, 96,331 (31%) had primary care data covering the full period between recruitment and 31 March 2017 or death, whichever came first.

## Polygenic risk scores (instrumental variables)

We used the Locke 2015 [30] genome-wide association study (GWAS) for BMI to identify genome-wide significant single nucleotide polymorphisms (SNPs) with strong evidence of association with BMI, defined as having a $P$ value below genome-wide significance ($P \leq 5 \times 10^{-8}$). We clumped the genome-wide significant SNPs at an $R^2$ threshold of 0.001 within a 10,000 kilobase window, and proxies were found for all SNPs not in UK Biobank using the European subsample of 1,000 genomes as a reference panel (with a lower $R^2$ limit of 0.6) [31]. In total, 69 SNPs were used to construct a PRS, which we calculated as the weighted sum of the SNP effect alleles for all SNPs associated with BMI, with each SNP weighted by the regression coefficient from the Locke GWAS. **S1 Table** shows summary data for all SNPs in the PRS. We did not use the more recent 2018 BMI GWAS because this includes the UK Biobank [32], and sample overlap leads to bias towards the observational effect in mendelian randomisation analyses [33].

## Exposure and covariates

We defined BMI as weight in kilograms divided by height in metres squared, and BMI categories using conventional World Health Organization guidelines [34]: normal weight as a BMI of between 18.5 kg/m$^2$ and 25 kg/m$^2$, overweight as a BMI of between 25 kg/m$^2$ and 30 kg/m$^2$, and obese as a BMI of above 30 kg/m$^2$. BMI was estimated at the UK Biobank baseline assessment using measured height and weight.

We used age, sex, and UK Biobank recruitment centre reported at the UK Biobank baseline assessment as covariables, as well as 40 genetic principal components derived by UK Biobank to control for population stratification [35].

## Data and code availability

This study is reported as per the Strengthening the Reporting of Observational Studies in Epidemiology (STROBE) guideline (**S1 STROBE Checklist**). This study did not have a prospective protocol or analysis plan: The analysis method was developed over the course of this study, and the policy analysis examples were considered before the method was finalised. No changes to the analysis were made from peer review comments. The empirical dataset is archived with UK Biobank and available to individuals who obtain the necessary permissions from the study's data access committees, with data accessible from https://www.ukbiobank.ac. uk/. The code used to clean and analyse the data is available here: https://github.com/sean-harrison-bristol/Robust-causal-inference-for-long-term-policy-decisions.

## Estimation of quality-adjusted life years and healthcare costs (outcomes)

**Quality-adjusted life years.** We predicted health-related quality of life for all participants daily from recruitment to 31 March 2017 using the results from a study by Sullivan and colleagues [36]; full details in **S3.1 Text**. Briefly, we used each of 240 chronic health conditions to predict health-related quality of life for all participants daily from recruitment to 31 March 2017 or death, whichever came first, and averaged over years to estimate QALYs. **S2 Table**

details all 240 chronic health conditions, including which ICD-9, ICD-10, read v2, and read v3 codes were used for each condition. QALYs are a measure of disease burden, capturing both the quality of life (through preferences over health states, which, in this context, may be understood as health-related quality of life) and quantity of life [37]. A QALY of 1 indicates a full year of perfect health, while a QALY of 0 indicates either a time of no quality of life or death. QALYs can be negative, implying that death would be preferable to life at a certain time. Throughout this manuscript, we report the change in the number of QALYs, either in whole numbers or percentage points, e.g., 0.65% of a QALY, meaning 0.0065 QALYs.

Chronic health conditions were recorded in an individual's primary care data, HES data, or both. As only 31% of participants in this study had primary care data, we used multiple imputation by chained equations to predict both QALYs and primary care healthcare costs (N missing = 214,270, 69%), creating 100 imputed datasets [38]. We also imputed Townsend deprivation index (N missing = 342, 0.1%) and whether the participant had ever smoked (N missing = 1,064, 0.3%), as these variables were informative but had some missingness. Further details are reported in **S3.2 Text**.

**Primary care healthcare costs.** We estimated primary care healthcare costs between recruitment and 31 March 2017 from the primary care data as the sum of the cost of prescribed drugs and appointments at a GP practice. Briefly, we estimated the cost of prescribed drugs during follow-up using the NHS electronic drug tariff (November 2019 version), adding the cost of each prescription (£1.27 in November 2019) to the cost of each drug [39]. In total, we costed 94% of 29,646,535 prescribed drugs, with the remaining drugs either no longer prescribed (and so not costed, *n* = 392,801, 1.3%) or unmatched to a price (*n* = 1,392,091, 4.7%). We estimated the cost of each appointment at a GP practice during follow-up at £30, an average of the cost of GP, nurse, and other appointments as we could not distinguish between consultation types from the available data [40]. We did not consider the cost of diagnostic tests. We divided the total primary care costs by years of follow-up to give the average yearly primary care healthcare costs for each participant.

**Secondary care healthcare costs.** We estimated secondary care healthcare (hospital) costs, in which we converted procedure and diagnosis ICD-10 codes from inpatient episodes into Healthcare Resource Groups, which are assigned a cost (in 2016/2017 pounds sterling) for publicly funded NHS hospitals; see Dixon (2019) for more information [41]. The data came from HES (for English care providers) and from the Patient Episode Database for Wales (for Welsh providers). Inpatients are those admitted to hospital and who occupy a hospital bed but need not necessarily stay overnight and does not include emergency care or outpatient appointments. We had follow-up data from baseline to 31 March 2015 for secondary care healthcare for all participants in this study. We estimated healthcare costs for those registered in England and Wales only, as the basis for remunerating hospitals in Scotland is different and cannot be combined with data from the other 2 countries [42].

We estimated the secondary care healthcare cost for each participant between recruitment and 31 March 2015, then divided by the years of follow-up to give the average secondary care healthcare cost per year of follow-up. Secondary care costs were therefore averaged over 2 fewer years than primary care costs. We increased the value of secondary care healthcare costs by 4.84% to reflect inflation between 2016/2017 and November 2019, using data from the NHS cost inflation index, with April to November 2019 inflation estimated at the average annual inflation in the previous 4 years accrued over 8 months [43].

**Total healthcare costs.** We combined the average yearly primary and secondary care healthcare costs for each person to estimate total NHS-based healthcare costs from inpatient hospital care episodes, primary care appointments, and primary care drug prescriptions. These costs exclude emergency care, outpatient appointments, and private healthcare undertaken in

private facilities (private healthcare received in NHS hospitals is included), in addition to diagnostic tests, but still represent a substantial proportion of healthcare costs in England and Wales. Including these other costs would likely increase the size of our effect estimate but would not alter the direction of the effect.

## Main analysis

We used mendelian randomisation to estimate the causal effect of BMI on QALYs and total healthcare costs per year using the PRS for BMI as an instrumental variable, with age at baseline assessment, sex, UK Biobank recruitment centre, and 40 genetic principal components as covariates. We used the ivreg2 package in Stata (version 15.1) with robust standard errors and tested for weak instrument bias (using F statistics) to assess whether the PRS for BMI was sufficiently associated with measured BMI [44]. This mendelian randomisation analysis estimates the mean difference in the outcomes using an additive structural mean model [45–47], interpreted as the average change in each outcome caused by a 1-kg/m$^2$ increase in BMI over all participants. We multiplied the results for QALYs by 100 to give the percentage of a QALY changed per unit increase in BMI.

**Comparison with multivariable regression approach.**   We compared the mendelian randomisation estimates with estimates from conventional multivariable linear regression for QALYs and healthcare costs, with age, sex, recruitment centre, and 40 genetic principal components as covariates. We performed endogeneity (Hausman) tests [48], in which a low $P$ value indicates that there was evidence the mendelian randomisation and multivariable effect estimates were different.

## Sensitivity analyses

S3.3 Text details full methods for all sensitivity analyses.

In brief, we conducted sensitivity analyses to test the mendelian randomisation assumption of no pleiotropy (i.e., that the genetic variants for BMI only affect each outcome through BMI) using summary data for each SNP in the BMI PRS, comprising inverse-variance weighted (IVW), MR Egger (an indicator of directional pleiotropy), weighted median, weighted mode, and simple mode analyses [49–51]. A low $P$ value in the MR Egger constant would indicate evidence of pleiotropy.

We also reran the main analysis stratified by age group (40 to 49, 50 to 54, 55 to 59, 60 to 64, and 65+ years) and by the World Health Organization BMI categories (normal weight, overweight, and obese) [34] to test and account for both nonlinearity and a potential interaction between age and BMI in the main effect estimates. We then used nonlinear mendelian randomisation to estimate the precise shape of the associations between BMI, QALYs, and healthcare costs [52,53]. Additionally, we conducted within-family mendelian randomisation to assess whether there was evidence that family structure biased estimates from the main analysis because nontransmitted genetic variants from parents may influence a child's individual healthcare costs and QALYs in later life [54,55].

We tested whether accounting for prediction uncertainty in QALYs made a material difference to the precision of the main analysis estimates of BMI on QALYs.

Finally, to test whether decision analytic simulation models incorporate enough health conditions to accurately estimate the effect of BMI on QALYs, we estimated whether including only limited health conditions (cancer, cardiovascular disease, cerebrovascular disease, and type 2 diabetes) in the prediction of QALYs had a substantial impact on the estimated effect of BMI on QALYs.

### Policy analyses

S3.4 Text details full methods for all policy analyses; S3.5 Text details a worked example of analysis **d**.

Briefly, we used the results from the mendelian randomisation analyses stratified by age and BMI categories, as well as data and parameter estimates from other studies, to estimate the effect of each of the following on QALYs and healthcare costs for the population aged 40 to 69 years of England and Wales in 2017 (21.7 million adults):

a. the effect of laparoscopic bariatric surgery in people with a BMI above 35 kg/m$^2$;

b. the effect of restricting volume promotions for HFSS foods;

c. the effect of the increase in BMI between 1993 and 2017; and

d. the effect of having the BMI profile of England and Wales in 2017 versus a hypothetical profile where no one has a BMI above 25 kg/m$^2$.

In example **a,** we estimated the net monetary benefit of laparoscopic bariatric surgery as compared to no intervention over 20 years at a cost-effectiveness threshold of £20,000 per QALY and a discount rate for both QALYs and costs of 3.5% per year. We estimated that there were 2,741,556 people (12.6%) aged 40 to 69 years with a BMI of 35 kg/m$^2$ or above in England and Wales in 2017. We assumed laparoscopic bariatric surgery reduced BMI by 25% (95% confidence interval [CI]: 22% to 28%) consistently over 20 years [56,57], and cost £9,549 [58]. In example **b,** we estimated the net monetary benefit of restricting volume promotions for HFSS foods as compared to no intervention over 1 year at a cost-effectiveness threshold of £20,000 per QALY. We assumed that the intervention reduced caloric intake by 11 to 14 calories per day, that weight is reduced by 0.042 kg per 1 fewer calorie consumed per day [59,60], and that the intervention had no cost. In example **c,** we estimated the change in QALYs and total healthcare costs each year for the change in BMI between 1993 and 2017, and in example **d,** we estimated the effect of overweight and obesity on QALYs and total healthcare costs each year. We estimated that there were 15,565,145 people (72%) in England and Wales in 2017 with a BMI above 25 kg/m$^2$.

We used data from the Health Survey for England in 1993 and 2017 to inform our estimates of the BMI distribution of people in England and Wales [1,2], and data from the Office of National Statistics to inform the age distribution in 2017 [61]. We defined the net monetary benefit as the change in QALYs due to the intervention multiplied by a cost-effectiveness threshold (£20,000), minus the change in healthcare costs due to the intervention and the cost of the intervention, including from complications for bariatric surgery for that particular intervention.

### Patient and public involvement

This study was conducted using UK Biobank. Details of patient and public involvement in the UK Biobank are available online (www.ukbiobank.ac.uk/about-biobank-uk/). No patients were specifically involved in setting the research question or the outcome measures, nor were they involved in developing plans for recruitment, design, or implementation of this study. No patients were asked to advise on interpretation or writing up of results. There are no specific plans to disseminate the results of the research to study participants, but the UK Biobank disseminates key findings from projects on its website.

## Results

In total, we included 310,913 unrelated white British participants from England and Wales in the analysis. These participants had a mean age of 56.9 years (standard deviation (SD) = 8.0

years), mean BMI of 27.4 kg/m$^2$ (SD = 4.8 kg/m$^2$), a mean follow-up time of 8.1 years (SD = 0.8 years) for primary care healthcare costs and HES data, a mean follow-up time of 6.1 years (SD = 0.8 years) for secondary care healthcare costs, and 10,519 participants died during follow-up (3.4%); see **Table 2**. The median QALY per person per year from the 100 imputed datasets was 0.78 (interquartile range (IQR) = 0.65 to 0.89), compared with 0.97 (IQR = 0.87 to 0.99) based on the HES data alone (nonimputed), reflecting incomplete information on chronic healthcare conditions in HES data. The median total healthcare cost per person per year was £601 (IQR = £212 to £1,217), the median primary care healthcare cost per year was £375 (IQR = £128 to £738), and the median secondary care healthcare cost per year was £88 (IQR = £0 to £494). All cost outcomes were positively skewed.

## Main analysis

We estimated in the mendelian randomisation analysis that a 1-kg/m$^2$ increase in BMI caused a reduction of 0.65% of a QALY per year (95% CI: 0.49% to 0.81%, *P* value = $1.2 \times 10^{-15}$) and a £42.23 increase in total healthcare costs per year (95% CI: £32.95 to £51.51, *P* value = $4.5 \times 10^{-19}$).

**Comparison with multivariable regression approach.** The multivariable adjusted analyses were consistent with the mendelian randomisation analyses, with median *P* values for endogeneity from imputed datasets 0.31 and 0.52 for QALYs and total healthcare costs respectively (**Table 3**). There was no evidence of weak instrument bias (the *F* statistic was 5,168). **Figs 2 and 3** show both the mendelian randomisation and multivariable adjusted estimates, for the main analysis, and stratified by sex, BMI category, and age category (see **Sensitivity analyses**).

## Sensitivity analyses

Full results from all sensitivity analyses are in **S4 Text**.

**Table 2. Summary demographics of UK Biobank.**

| Variable | All | Men | Women |
|---|---|---|---|
| N | 310,913 | 144,032 | 166,881 |
| Age at recruitment, years [Mean (SD)] | 56.9 (7.99) | 57.1 (8.10) | 56.7 (7.90) |
| BMI, kg/m$^2$ [Mean (SD)] | 27.4 (4.75) | 27.8 (4.22) | 27.0 (5.13) |
| Years of follow-up [Mean (SD)] | 8.1 (0.80) | 8.1 (0.80) | 8.1 (0.80) |
| Participants with complete primary care data [N (%)] | 96,331 (30.98) | 44,671 (31.01) | 51,660 (30.96) |
| Death before 31 March 2017 [N (%)] | 10,519 (3.38) | 6,447 (4.48) | 4,072 (2.44) |
| Qualification: None [N (%)] | 54,874 (17.65) | 25,340 (17.59) | 29,534 (17.70) |
| Qualification: A levels, O level, GCSE, or CSE [N (%)] | 122,971 (39.55) | 51,475 (35.74) | 71,496 (42.84) |
| Qualification: NVQ or other [N (%)] | 36,288 (11.67) | 19,873 (13.80) | 16,415 (9.84) |
| Qualification: College or university degree [N (%)] | 96,780 (31.13) | 47,344 (32.87) | 49,436 (29.62) |
| Average QALYs per year (predicted) [Median (IQR)]* | 0.78 (0.65 to 0.89) | 0.78 (0.65 to 0.89) | 0.78 (0.65 to 0.88) |
| Annual total healthcare costs [Median (IQR)]* | £601 (£212 to £1,217) | £605 (£206 to £1,240) | £596 (£216 to £1,199) |

*Results from imputed data, median, and IQR are the medians of the 100 imputed medians/IQRs.

BMI, body mass index; IQR, interquartile range; *N*, number of participants; QALYs, quality-adjusted life years; SD, standard deviation.

**Table 3. Results from the main mendelian randomisation analysis.**

| Outcome | Main MR Analysis | | Multivariable Adjusted Analysis | | P value for Endogeneity |
|---|---|---|---|---|---|
| | Beta (95% CI) | P value | Beta (95% CI) | P value | |
| QALYs per year | −0.65% (−0.81% to −0.49%) | $1.2 \times 10^{-15}$ | −0.71% (−0.73% to −0.69%) | $<1 \times 10^{-323}$ | 0.31 |
| Total healthcare costs per year | £42.23 (£32.95 to £51.51) | $4.5 \times 10^{-19}$ | £39.40 (£38.19 to £40.61) | $<1 \times 10^{-323}$ | 0.52 |

Both analyses adjusted for age, sex, recruitment centre, and 40 genetic principal components.

Beta, effect estimate (beta coefficient) from analysis; CI, confidence interval; MR, mendelian randomisation; QALYs, quality-adjusted life years.

Results for QALYs are expressed as percentage points, e.g., 0.65% is equivalent to 0.0065 QALYs.

Briefly, from the summary mendelian randomisation sensitivity analyses, we found little evidence of pleiotropy in the mendelian randomisation estimates, but evidence of heterogeneity in SNP effects using Cochran's Q value (**S3 Table**).

We found little difference between the effect estimates when analysing men and women separately; **S1**–**S19 Tables** have results split by sex. However, we found strong evidence of nonlinearity in the effect of BMI on QALYs, where the effect of the same increase in BMI on QALYs was higher in overweight and obese participants than normal weight participants. There was little evidence of the same nonlinearity for total healthcare costs, although this may be due to a lack of power to detect the effects; see **Figs 4 and 5** and **S4 and S7 Tables**. Additionally, we found evidence for an interaction between BMI and age for both QALYs and total healthcare costs, where the effect of a unit increase in BMI increased as age increased (**S5 Table**). These results indicate that accounting for sex is not necessary when applying these results to cost-effectiveness analyses, but accounting for age and nonlinearity of the BMI effect is necessary.

The within-family mendelian randomisation analysis estimate for QALYs was very similar to the main analysis estimate but was smaller for total healthcare costs, though both estimates were far less precise (**S8 Table**). Accounting for the uncertainty in the QALY predictions increased the standard errors of both effect estimates, but not substantially, and did not change the effect estimates (**S9 Table**).

Predicting QALYs using a limited number of health conditions, as is often done in decision analytic simulation models, drastically reduced the estimated effect of BMI on QALYs, from −0.65% of a QALY (95% CI: −0.49% to −0.81%) to a reduction of 0.16% of a QALY (95% CI: 0.10% to 0.22%) per 1-kg/m$^2$ increase in BMI. This indicates that BMI affects more health conditions than just cancer, cardiovascular disease, cerebrovascular disease, and type 2 diabetes, and these other conditions have a considerable impact on health-related quality of life (**S10 Table**).

## Policy analyses

**Cost-effectiveness of laparoscopic bariatric surgery.** We estimated that 2,741,556 people in England and Wales had a BMI above 35 kg/m$^2$ in 2017. Compared to no intervention, over 20 years for each person receiving laparoscopic bariatric surgery we estimated that QALYs would increase by 0.92 (95% CI: 0.66 to 1.17), total healthcare costs would decrease by £5,096 (95% CI: £3,459 to £6,852), and the net monetary benefit (at £20,000 per QALY and £9,549 per intervention) would be £13,936 (95% CI: £8,112 to £20,658). Therefore, laparoscopic bariatric surgery is very likely to be cost-effective over 20 years for people with BMI of 35 kg/m$^2$ aged 40 to 69 years in England and Wales. Multivariable adjusted estimates were larger for QALYs and similar for costs, both with greater precision. Full results are in **S11** and **S12 Tables**.

**Cost-effectiveness of restricting volume promotions for high fat, sugar, and salt (HFSS) products.** We estimated that restricting volume promotions for HFSS products would, across 21 million adults in England and Wales, increase QALYs by 20,551 per year (95% CI: 15,335 to

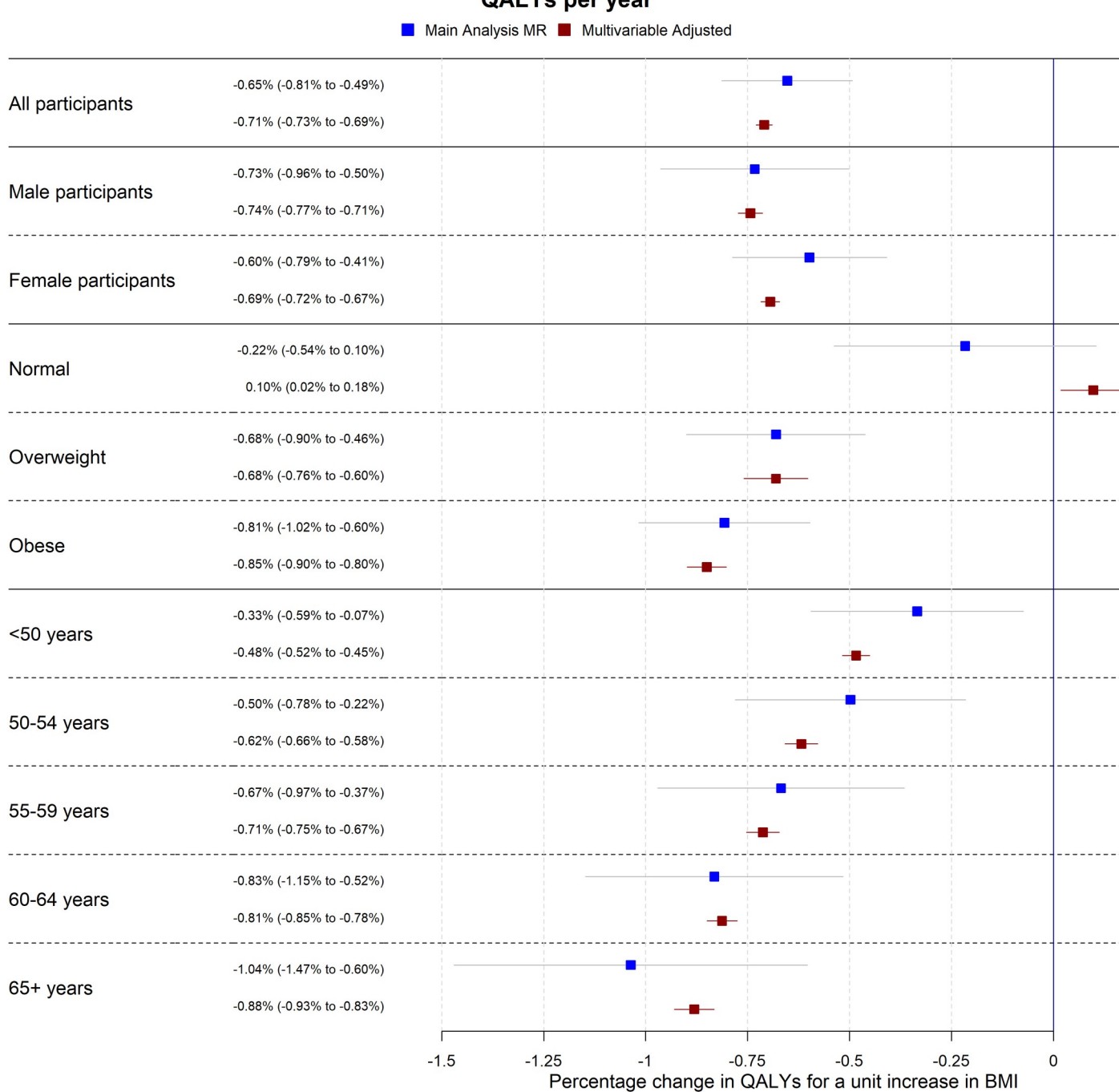

**Fig 2. MR estimates for QALYs per year.** Forest plot showing the estimated effect of a unit increase in BMI on average QALYs per year for the main MR, sex-specific, BMI categorical (where "Normal" is a BMI below 25 kg/m², "Overweight" is a BMI between 25 kg/m² and 30 kg/m², and "Obese" is a BMI of above 30 kg/m²) and age categorical analyses. Effect estimates are indicated by squares, 95% CIs by horizontal lines around the squares. Effect estimates are derived from the main imputation model (for all and sex-specific estimates) or the categorical imputation model (for BMI and age category–specific estimates). Both analyses adjusted for age, sex, recruitment centre, and 40 genetic principal components. BMI, body mass index; CI, confidence interval; MR, mendelian randomisation; QALY, quality-adjusted life year.

25,301), decrease total healthcare costs by £137 million per year (95% CI: £106 million to £170 million), and would have a net monetary benefit (at £20,000 per QALY and no intervention

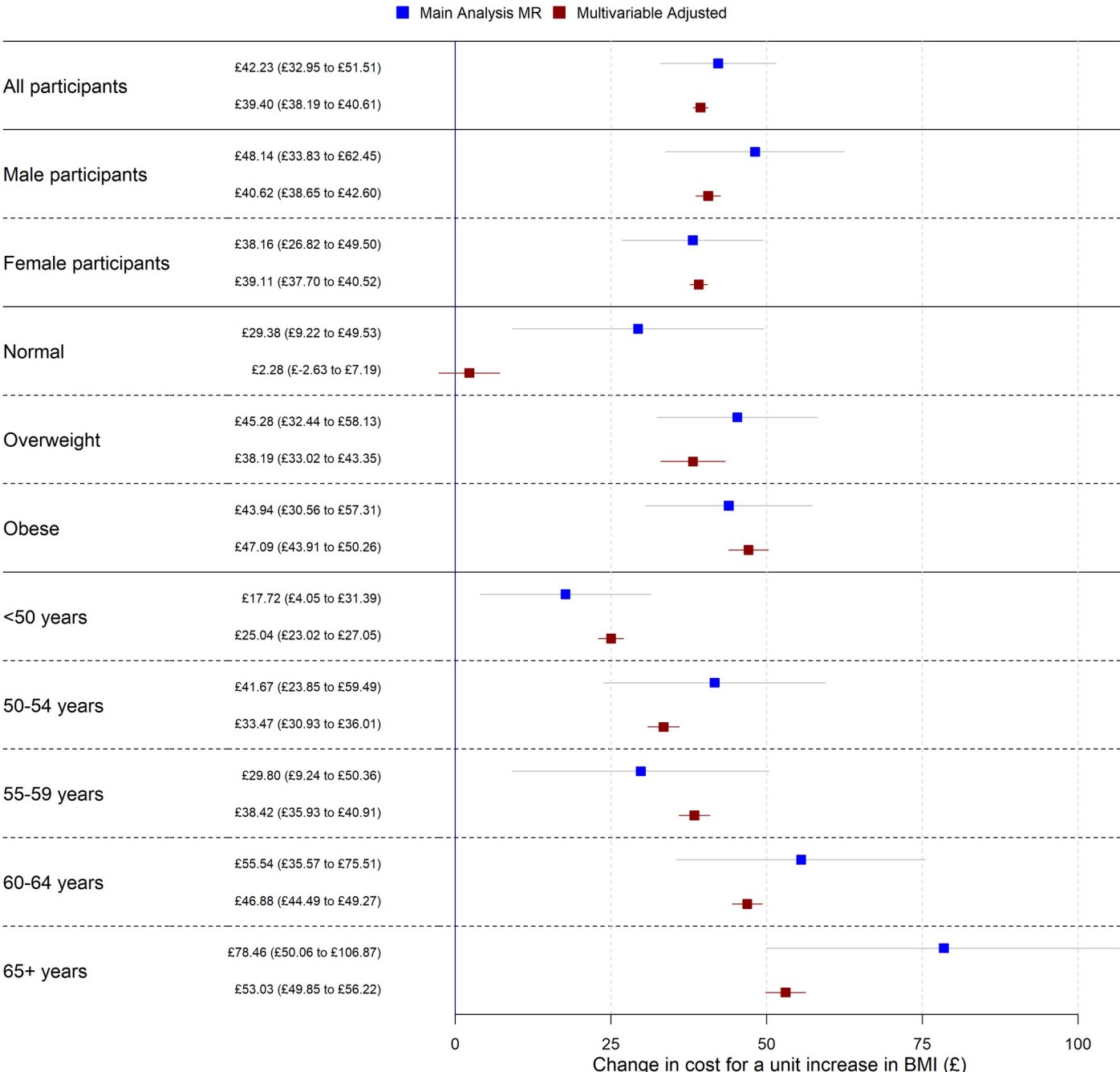

**Fig 3. MR estimates for total healthcare costs per year.** Forest plot showing the estimated effect of a unit increase in BMI on average total healthcare costs per year for the main MR, sex-specific, BMI categorical (where "Normal" is a BMI below 25 kg/m$^2$, "Overweight" is a BMI between 25 kg/m$^2$ and 30 kg/m$^2$, and "Obese" is a BMI of above 30 kg/m$^2$) and age categorical analyses. Effect estimates are indicated by squares, 95% CIs by horizontal lines around the squares. Effect estimates are derived from the main imputation model (for all and sex-specific estimates) or the categorical imputation model (for BMI and age category–specific estimates). Both analyses adjusted for age, sex, recruitment centre, and 40 genetic principal components. BMI, body mass index; CI, confidence interval; MR, mendelian randomisation.

cost) of £546 million per year (95% CI: £435 million to £671 million). The intervention would therefore almost certainly be cost effective, relative to doing nothing. Multivariable adjusted

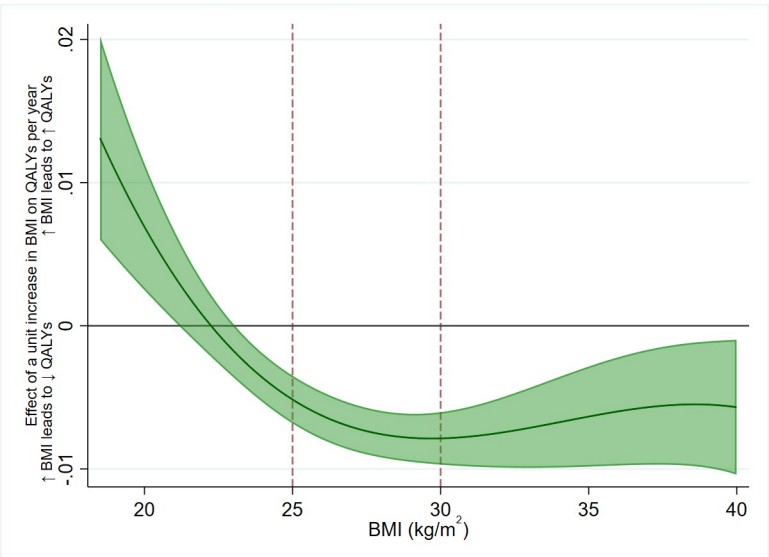

**Fig 4. The estimated effect of 1-kg/m² increase in BMI on QALYs per year, across BMI levels.** A positive value indicates an increase in BMI would increase QALYs, and vice versa. An increase in BMI is beneficial to QALYs up to around 22 kg/m², then becomes increasingly detrimental until the effect plateaus in overweight and remains relatively steady in obesity. The BMI thresholds of 25 kg/m² (overweight) and 30 kg/m² (obese) are represented with dashed red lines. The green shaded area represents the 95% CI of the estimated effect. Effect estimates are derived from the nonlinear imputation model. BMI, body mass index; CI, confidence interval; QALY, quality-adjusted life year.

estimates were larger for QALYs and similar for costs, both with greater precision. Full results are in **S13** and **S14 Tables**.

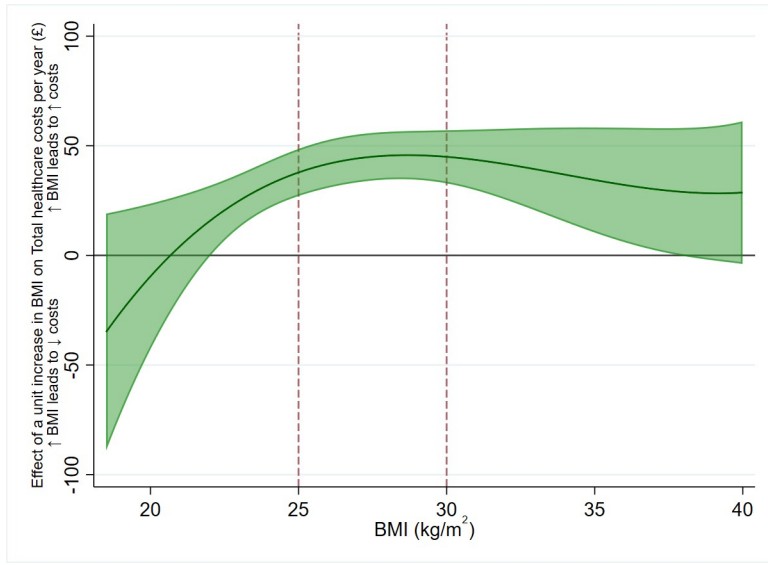

**Fig 5. The effect of 1-kg/m² increase in BMI on total healthcare costs per year, across BMI levels.** A positive value indicates that an increase in BMI would increase total healthcare costs, and vice versa. Due to the uncertainty in the estimates, there is little statistical evidence of nonlinearity in the effect of BMI on total healthcare costs, though descriptively, it appears that a 1-kg/m² increase in BMI has a smaller effect on costs in the normal weight category, and a larger effect in overweight and obesity. The BMI thresholds of 25 kg/m² (overweight) and 30 kg/m² (obese) are represented with dashed red lines. The green shaded area represents the 95% CI of the estimated effect. Effect estimates are derived from the nonlinear imputation model. BMI, body mass index; CI, confidence interval.

**Estimation of the effect of the population change in BMI between 1993 and 2017.**
Mean BMI increased from 26.7 kg/m$^2$ to 28.6 kg/m$^2$ between 1993 and 2017 in people aged
between 40 and 69 years in England and Wales. The rise in BMI was more pronounced in people with obesity than people with a normal weight; see **S15 Table**.

We estimated that between 1993 and 2017, across 21 million adults in England and Wales,
the increase in BMI led to an average decrease in QALYs of 1.13% of a QALY per person per
year (95% CI: 0.90% to 1.38%), or a decrease of 246,390 QALYs in total per year (95% CI:
196,231 to 300,481) and an increase in total healthcare costs of £69 per person per year (95%
CI: £53 to £84), or £1.50 billion in total per year (95% CI: £1.15 billion to £1.82 billion), giving
a combined cost (at £20,000 per QALY) of £312 per person per year (95% CI: £235 to £347), or
£6.39 billion (95% CI: £5.12 billion to £7.54 billion). This indicates that an intervention, which
could reduce the BMI of the population of England and Wales to 1993 levels, would likely be
cost effective if it cost less than £5.12 billion per year. Multivariable adjusted estimates were
larger for QALYs and similar for costs, both with greater precision. Full results are in **S16** and
**S17 Tables**.

**The cost of being overweight and obese in 2017.**   We estimated that, compared to if all
people with a BMI above 25 kg/m$^2$ aged 40 to 69 years in England and Wales in 2017 had a
BMI of 25 kg/m$^2$, the current BMI profile of England and Wales decreases QALYs by 3.73% of
a QALY per person with a BMI above 25 kg/m$^2$ per year (95% CI: 2.94% to 4.61%), or a
decrease of 580,494 QALYs in total per year (95% CI: 457,907 to 717,691), and increases total
healthcare costs by £230 per person per year (95% CI: £176 to £279), or £3.58 billion in total
per year (95% CI: £2.75 billion to £4.34 billion), giving a combined cost (at £20,000 per QALY)
of £973 per person per year (95% CI: £773 to £1160), or £15.1 billion (95% CI: £12.0 billion to
£18.1 billion). Multivariable adjusted estimates were larger for QALYs and similar for costs,
both with greater precision. Full results are in **S18** and **S19 Tables**.

## Discussion

In this study, we have shown that cost-effectiveness of clinical and policy interventions can be
estimated using mendelian randomisation. We estimated the effect of a unit increase in BMI
on average QALYs and total healthcare costs per year in UK Biobank, which showed that
increasing BMI is detrimental to both QALYs and healthcare costs. The effect of an increase
BMI on healthcare costs and QALYs was relatively stable for BMI values above 25 kg/m$^2$,
implying that the expected effect of a change in BMI is very similar whether a person has a
BMI considered overweight or obese. We used these estimates to show that bariatric surgery
and the restriction of volume promotions for HFSS products are likely cost-effective relative to
a "no intervention" comparator (net monetary benefit of £13,936 over 20 years) and estimated
the costs of the increase to BMI over time (a decrease of 1.13% of a QALY and increase of £69
of annual healthcare costs per person) and having a BMI above 25 kg/m$^2$ in 2017 (a decrease of
3.73% of a QALY and increase of £230 of annual healthcare costs per person).

We have demonstrated how mendelian randomisation can be useful for estimating the
impact on quality of life and healthcare costs of either an exposure or intervention that is difficult, unethical, or impossible to randomise (e.g., smoking, alcohol intake), or for interventions
where long-term cost-effectiveness evidence from RCTs is rare or not generalisable (e.g., bariatric surgery). While in this study the conventional multivariable adjusted estimates not using
genetic information were mostly similar to the mendelian randomisation estimates, this could
be due to larger uncertainty in the mendelian randomisation estimates, and there is no guarantee that other exposures will be similar. We have also shown that considering more health conditions than cancer, cardiovascular disease, cerebrovascular disease, and type 2 diabetes

considerably increases the estimated effect of BMI on QALYs and healthcare costs, that laparoscopic bariatric surgery is likely to be cost-effective, and that the costs of population-level changes in BMI can be substantial.

Previous studies examining the cost-effectiveness of interventions for obesity have used RCTs [9], cohorts [10–13], and decision analytic and related simulation models [10,12,14–18]. These studies estimated the impact on QALYs and the total healthcare cost of different interventions, such as bariatric surgery, and thus estimated whether the intervention was likely to be cost-effective. Relative to existing methods, mendelian randomisation has longer follow-up, is less expensive and quicker, combines a more comprehensive set of outcomes, and is less likely to suffer from confounding and reverse causation. However, the disadvantages to mendelian randomisation for cost-effectiveness analysis are that it requires larger sample sizes, and we cannot be certain that the effects of lifelong changes in BMI due to genetics will be comparable to changes induced by interventions. These relative strengths and limitations of the different approaches are summarised in **Table 1**.

## Strengths and limitations

The estimates of the effect of BMI on QALYs and costs from mendelian randomisation are likely less biased by confounding and reverse causation than either cohort studies or decision analytic simulation models using observational effect estimates [20]. UK Biobank has many participants with comprehensive information about costs and disease states over many years. While the corresponding conventional multivariable adjusted estimates were generally consistent with the mendelian randomisation estimates for all outcomes, the mendelian randomisation estimates showed some detrimental effect of increasing BMI even in participants with BMI close to the top end of the normal weight category, while the conventional estimates did not, which could reflect bias in the conventional estimates.

This method of estimating the effect of a risk factor on QALYs and costs can be extended to other risk factors with causal genetic components and also provide evidence for the causal effects of health conditions on healthcare costs and QALYs. This may be useful for health conditions that are strongly influenced by risk factors that affect other health conditions where the effect of the condition would otherwise be confounded by the risk factor, such as cardiovascular disease.

However, mendelian randomisation relies on assumptions that cannot be proven [20], as is the case with all types of instrumental variable analysis and other forms of observational policy evaluation. There was evidence for heterogeneity between SNPs for all outcomes, though in general, the summary mendelian randomisation sensitivity estimates were consistent with the main estimates, and there was little evidence of directional pleiotropy from the MR Egger regression. As the outcomes were not biological, the exclusion restriction assumption (i.e., that any genetic variant affects the outcome only through the exposure) may not hold for all the genetic variants (i.e., that the genetic variant affects the outcome only through the exposure).

These estimates represent a lifetime exposure to a genetic influence on BMI and thus cannot be interpreted directly as the expected effect of an intervention at a specific age. In general, as the age at which a person received an intervention increases, the effect estimates would likely reduce. This is because the mechanisms by which BMI affects health may be cumulative over time, and so even if BMI were lowered in older age, some residual detrimental effect of previously high BMI may remain. It is therefore likely that our estimates of the impact of BMI on costs and QALYs are best applied to population level interventions that aim to reduce BMI across all age groups. This limitation is also present in decision analytic simulation models of cost-effectiveness, though not RCTs or cohort studies. Our estimates may also underestimate

the true effect as people in England and Wales now may have had larger BMI values earlier in life than previously, increasing the length of exposure to obesity. It is also the case that the mendelian randomisation estimates may be fully representative of interventions that target BMI, as these interventions will typically target more than just a change in BMI, including exercising more or improving diets. Therefore, the generalisability of our results to interventions for BMI will depend on how comparable the intervention is to causing a genetically determined difference in BMI.

For all policy examples, we require the stable unit treatment value assumption for causal inference; this assumption requires that genetic change in BMI is equivalent to a change in BMI by other means, e.g., by bariatric surgery or reducing caloric intake of HFSS foods. This assumption is not testable. Mendelian randomisation analyses can also be interpreted as estimates of a "local average treatment effect," by assuming that changes in the genetic variants affecting BMI affect all participants in UK Biobank in the same direction (monotonicity). This assumption also cannot be tested, and deviations from monotonicity could bias effect estimates.

The analyses accounting for QALY prediction error were consistent with the main analysis, although less precise. We predicted QALYs using data from Sullivan and colleagues [36], as QALYs have not been previously estimated in UK Biobank. While these data are applicable to a UK population, this method only captures health-related quality of life, and, therefore, our QALY estimates do not include any non-health-related determinants of quality of life. This was unavoidable given the data available in UK Biobank, where only linked healthcare data were available beyond baseline (excepting the relatively small amount of data from follow-up visits): Future studies repeatedly measuring quality of life directly may therefore provide more robust effect estimates. We also had to impute primary care costs and QALYs as only a limited section of UK Biobank had primary care data, which limited statistical power but were unlikely to have biased the results; rather, the complete case analysis would likely have been biased results, since the distribution of GP software systems allowing linkage of primary care data is unlikely to be random.

The healthcare costs were estimated from observed hospital episodes, drug prescriptions, and appointments from primary care. Follow-up was 2 years shorter for secondary care costs than primary care costs, but as we averaged the costs, this should not have materially affected the results. Additionally, we did not capture all healthcare costs as we did not have access to private healthcare costs not incurred in NHS settings, or data for emergency care or outpatient appointments (which are not linked to the UK Biobank cohort), and did not consider the cost of diagnostic tests in primary care, likely therefore underestimating the total cost of increasing BMI. In contrast, participants in UK Biobank may have different access to healthcare than the country on average, which may have biased our estimates of the effect of BMI on costs. Finally, BMI may have interacted with the use of both state and private healthcare, potentially biasing the results in either direction.

In the policy analyses, we made several assumptions: that bariatric surgery had no effects on QALYs through anything other than its effect on BMI, including no perioperative mortality or side effects (though complications of bariatric surgery on total healthcare costs up to 5 years were included in the cost of surgery); that the estimated BMI reduction from bariatric surgery would be maintained over 20 years; and that both UK Biobank and the Health Survey for England were representative of the population of England and Wales. These assumptions appear justifiable, as the average effect of bariatric surgery on QALYs over 20 years is likely relatively low, bariatric surgery has shown a consistent reduction in BMI up to 20 years [56,57], and the Health Survey for England is nationally representative [1,2].

However, despite its size, UK Biobank is not representative of the UK population as participants tend to be wealthier and healthier compared to the country on average [62]. It therefore

likely that we have underestimated the true costs of BMI, as wealthier and healthier people may be more resistant to any detrimental effects of increased BMI. As obesity is more common in lower socioeconomic groups [63], our results suggest that obesity may be causally related to inequalities in quality of life.

Although mendelian randomisation is likely to be less affected by confounding and reverse causality than conventional multivariable adjusted analyses, an important potential source of bias in these analyses is family-level effects. Recent evidence suggests that assortative mating and dynastic effects can lead to bias in mendelian randomisation effect estimates [54], though within-family mendelian randomisation studies can account for some of these biases. Our within-family sensitivity analyses showed that the effect of BMI on QALYs was consistent with the main analysis, though the effect of BMI on total healthcare costs was reduced. However, statistical power was limited in these analyses, and confidence intervals were wide. Additionally, there is evidence of a geographic structure in the UK Biobank genotype data that cannot be accounted for using adjustment for principal components, which may also have biased our analyses [64].

## Conclusions

Mendelian randomisation can be used to estimate the effect of an exposure on quality of life and healthcare costs. We used this approach to estimate the cost-effectiveness of interventions aimed at reducing BMI, all of which we estimated were likely to be cost-effective, and found that the effect of increasing BMI on health-related quality of life may be larger than previously thought, as decision analytic simulation models may underestimate the effect of BMI on QALYs by using only limited health conditions are intermediates.

This approach could be especially useful where it is difficult, unethical, or impossible to randomise participants to an exposure such as obesity or for prevalent behaviours with adverse health impacts such as smoking or alcohol use, or where RCT evidence is rare for an intervention. Results from such studies are likely of benefit to both policy and the NHS. In future studies, we will use this method to assess the costs of different risk factors for poor health.

## Supporting information

**S1 Text. Description of studies estimating the cost-effectiveness of interventions.**
(DOCX)

**S2 Text. Inclusion criteria and genotyping. Fig A in S2 Text.** Flow chart for study inclusion/ exclusion.
(DOCX)

**S3 Text.** Supplementary methods, including 3.1: Estimation of health-related quality of life, 3.2: Dealing with missing data, 3.3: Sensitivity analyses, 3.4: Policy analyses, and 3.5: Worked example of a policy analysis.
(DOCX)

**S4 Text. Sensitivity analyses: Results. Fig A in S4 Text.** The estimated effect of a $1\text{-kg/m}^2$ increase in BMI on average QALYs per year for each quantile of PRS-free BMI. The solid green line indicates the trend line using cubic variance-weighted least squares. The dashed navy lines indicate the PRS-free BMI category specific estimates from the main mendelian randomisation analysis. The effect estimate for each quantile and its 95% CI is represented by the blue points and red vertical lines. BMI, body mass index; CI, confidence interval; PRS, polygenic risk score; QALY, quality-adjusted life year. **Fig B in S4 Text**. The estimated effect of a $1\text{-kg/m}^2$ increase in BMI on average total healthcare cost per year for each quantile of PRS-free

BMI. The solid green line indicates the trend line using cubic variance-weighted least squares. The dashed navy lines indicate the PRS-free BMI category specific estimates from the main mendelian randomisation analysis. The effect estimate for each quantile and its 95% CI is represented by the blue points and red vertical lines. BMI, body mass index; CI, confidence interval; PRS, polygenic risk score; QALY, quality-adjusted life year.
(DOCX)

**S1 STROBE Checklist. STROBE Checklist.**
(DOCX)

**S1 Table. SNPs used in the BMI PRS.**
(XLSX)

**S2 Table. Medical condition HES and primary care codes.** All ICD 9, ICD 10, Read 2, and Read 3 codes used to code the 240 included medical conditions in UK Biobank HES and primary care data. Codes and medical conditions used in **sensitivity analysis f** are listed (cancer, cardiovascular disease, cerebrovascular disease, and type 2 diabetes).
(XLSX)

**S3 Table. Summary MR analysis results (sensitivity analysis a).**
(XLSX)

**S4 Table. Main mendelian randomisation and multivariable adjusted analysis results, including results by sex, and age and BMI categories (sensitivity analyses b and c).**
(XLSX)

**S5 Table. Age interaction mendelian randomisation and multivariable adjusted analysis results (sensitivity analysis b).**
(XLSX)

**S6 Table. BMI quantile results: Results from the PRS-free BMI quantile mendelian randomisation and multivariable adjusted analyses used to inform the nonlinear analyses (sensitivity analysis d).**
(XLSX)

**S7 Table. Nonlinear BMI results: Results from the nonlinear mendelian randomisation and multivariable adjusted analyses (sensitivity analysis d).**
(XLSX)

**S8 Table. Within-family mendelian randomisation and multivariable adjusted analysis results (sensitivity analysis e).**
(XLSX)

**S9 Table. Results from mendelian randomisation and multivariable adjusted analyses accounting for uncertainty in the QALY predictions, both accounting for death and not accounting for death (sensitivity analysis f).**
(XLSX)

**S10 Table. Results from mendelian randomisation and multivariable adjusted analyses only including limited health conditions (cancer, cardiovascular disease, cerebrovascular disease, and type 2 diabetes) in the estimation of QALYs (sensitivity analysis g).**
(XLSX)

**S11 Table. Results for the cost effectiveness of laparoscopic bariatric surgery (total population of 2,741,556 people in England and Wales with a BMI of above 35 kg/m$^2$) using**

**mendelian randomisation and multivariable adjusted estimates (Policy Analysis a).**
(XLSX)

**S12 Table. Results for the cost-effectiveness of laparoscopic bariatric surgery (per person in England and Wales with a BMI of above 35 kg/m$^2$) using mendelian randomisation and multivariable adjusted estimates (Policy Analysis a).**
(XLSX)

**S13 Table. Results for the cost-effectiveness of restricting volume promotions on high fat, salt, and sugar products (total population of 21,742,497 people aged 40 to 69 years in England and Wales) using mendelian randomisation and multivariable adjusted estimates (Policy Analysis b).**
(XLSX)

**S14 Table. Results for the cost-effectiveness of restricting volume promotions on high fat, salt, and sugar products (per person aged 40 to 69 years in England and Wales) using mendelian randomisation and multivariable adjusted estimates (Policy Analysis b).**
(XLSX)

**S15 Table. Estimates of the change in mean BMI between 1993 and 2017 using data from the Health Survey for England.**
(XLSX)

**S16 Table. Results for the estimation of the effect of the population change in BMI between 1993 and 2017 (total population of 21,742,497 people aged 40 to 69 years in England and Wales) using mendelian randomisation and multivariable adjusted estimates (Policy Analysis c).**
(XLSX)

**S17 Table. Results for the estimation of the effect of the population change in BMI between 1993 and 2017 (per person aged 40 to 69 years in England and Wales) using mendelian randomisation and multivariable adjusted estimates (Policy Analysis c).**
(XLSX)

**S18 Table. Results for the estimation of the cost of being overweight (BMI > 25 kg/m$^2$) (total population of 21,742,497 people aged 40 to 69 years in England and Wales) using mendelian randomisation and multivariable adjusted estimates (Policy Analysis d).**
(XLSX)

**S19 Table. Results for the estimation of the cost of being overweight (BMI > 25 kg/m$^2$) (per person aged 40 to 69 years in England and Wales) using mendelian randomisation and multivariable adjusted estimates (Policy Analysis d).**
(XLSX)

## Acknowledgments

This research has been conducted using the UK Biobank Resource under Application Number 29294. Quality Control filtering of the UK Biobank data was conducted by R. Mitchell, G. Hemani, T. Dudding, and L. Paternoster as described in the published protocol (doi: 10.5523/bris.3074krb6t2frj29yh2b03x3wxj). The MRC IEU UK Biobank GWAS pipeline was developed by B. Elsworth, R.Mitchell, C. Raistrick, L. Paternoster, G. Hemani, and T. Gaunt (doi: 10.5523/bris.pnoat8cxo0u52p6ynfaekeigi).

This publication is the work of the authors, who serve as the guarantors for the contents of this paper.

The lead author (the manuscript's guarantor) affirms that this manuscript is an honest, accurate, and transparent account of the study being reported; that no important aspects of the study have been omitted; and that any discrepancies from the study as planned (and, if relevant, registered) have been explained.

## Author Contributions

**Conceptualization:** Sean Harrison, Padraig Dixon, Hayley E. Jones, Laura D. Howe, Neil M. Davies.

**Data curation:** Sean Harrison.

**Formal analysis:** Sean Harrison.

**Funding acquisition:** Hayley E. Jones, Alisha R. Davies, Laura D. Howe, Neil M. Davies.

**Methodology:** Sean Harrison, Padraig Dixon, Hayley E. Jones, Laura D. Howe, Neil M. Davies.

**Software:** Sean Harrison, Padraig Dixon.

**Supervision:** Hayley E. Jones, Neil M. Davies.

**Visualization:** Sean Harrison.

**Writing – original draft:** Sean Harrison.

**Writing – review & editing:** Sean Harrison, Padraig Dixon, Hayley E. Jones, Alisha R. Davies, Laura D. Howe, Neil M. Davies.

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
