## [Editor Report · Decision Letter 0]

27 May 2020

Dear Dr Harrison, 

Thank you for submitting your manuscript entitled "Robust causal inference for long-term policy decisions: cost effectiveness of interventions for obesity using Mendelian randomization" for consideration by PLOS Medicine.

Your manuscript has now been evaluated by the PLOS Medicine editorial staff and I am writing to let you know that we would like to send your submission out for external peer review.

Kind regards,

Helen Howard, for Clare Stone PhD 

Acting Editor-in-Chief

PLOS Medicine 

plosmedicine.org

---

## [Decision Letter · Decision Letter 1]

9 Apr 2021

Dear Dr. Harrison,

Thank you very much for submitting your manuscript "Robust causal inference for long-term policy decisions: cost effectiveness of interventions for obesity using Mendelian randomization" (PMEDICINE-D-20-02167R1) for consideration at PLOS Medicine. 

Your paper was evaluated by a senior editor and discussed among all the editors here. It was also sent to four independent reviewers, including a statistical reviewer. The reviews are appended at the bottom of this email and any accompanying reviewer attachments can be seen via the link below:

[LINK]

In light of these reviews, I am afraid that we will not be able to accept the manuscript for publication in the journal in its current form, but we would like to consider a revised version that addresses the reviewers' and editors' comments. Obviously we cannot make any decision about publication until we have seen the revised manuscript and your response, and we plan to seek re-review by one or more of the reviewers. 

We expect to receive your revised manuscript by Apr 30 2021 11:59PM. Please email us (plosmedicine@plos.org) if you have any questions or concerns.

We look forward to receiving your revised manuscript. 

Sincerely,

Caitlin Moyer, PhD

Associate Editor 

PLOS Medicine

plosmedicine.org

1. Data availability statement: Please provide the links (Github or other) for the code underlying the study. Please provide the access links for the UK Biobank data.

2. Financial disclosure: Please slightly revise the final sentence of this section to read “The funders had no role in study design, data collection and analysis, decision to publish, or preparation of the manuscript”

3. Title: Please revise your title according to PLOS Medicine's style. Your title must be nondeclarative and not a question. It should begin with main concept if possible. "Effect of" should be used only if causality can be inferred, i.e., for an RCT. Please place the study design ("A randomized controlled trial," "A retrospective study," "A modelling study," etc.) in the subtitle (ie, after a colon).

4. Abstract: Please structure your abstract using the PLOS Medicine headings (Background, Methods and Findings, Conclusions).

5. Abstract Background: Provide the context of why the study is important. The final sentence should clearly state the study question.

6. Abstract: Please clarify this phrase: “When considering only health conditions usually considered in previous studies...”

7. Abstract: Methods and Findings: Please provide more details of the study design and the main outcome measures before presenting the findings.

8. Abstract: Methods and Findings: In the last sentence of the Abstract Methods and Findings section, please describe the main limitation(s) of the study's methodology.

9. Abstract: Conclusions: Please address the study implications without overreaching what can be concluded from the data; the phrase "In this study, we observed ..." may be useful.

10. Author Summary: At this stage, we ask that you include a short, non-technical Author Summary of your research to make findings accessible to a wide audience that includes both scientists and non-scientists. The Author Summary should immediately follow the Abstract in your revised manuscript. This text is subject to editorial change and should be distinct from the scientific abstract. Please see our author guidelines for more information: https://journals.plos.org/plosmedicine/s/revising-your-manuscript#loc-author-summary

11. Throughout: Please use square brackets for in-text citations. Please include line numbers with your revised document.

12. Methods: Please ensure that the study is reported according to the STROBE guideline, and include the completed STROBE checklist (or the most appropriate checklist for your study) as Supporting Information. When completing the checklist, please use section and paragraph numbers, rather than page numbers. Please add the following statement, or similar, to the Methods: "This study is reported as per the Strengthening the Reporting of Observational Studies in Epidemiology (STROBE) guideline (S1 Checklist)."

13. Methods: Did your study have a prospective protocol or analysis plan? Please state this (either way) early in the Methods section.

14. Results: Please present results with 95% CIs and p values for all applicable analyses.

15. Discussion: Please present and organize the Discussion as follows: a short, clear summary of the article's findings; what the study adds to existing research and where and why the results may differ from previous research; strengths and limitations of the study; implications and next steps for research, clinical practice, and/or public policy; one-paragraph conclusion.

16. References: Please use the "Vancouver" style for reference formatting, and see our website for other reference guidelines https://journals.plos.org/plosmedicine/s/submission-guidelines#loc-references

17. Figures and Tables: Please provide titles and legends for all figures and tables (including those in Supporting Information files).

18. Figures 2 and 3: Please note in the legend these points are shown with 95% CIs.

19. Figure 4 and 5: Please note the dotted vertical lines in the legend. Please note the shaded area indicates the confidence interval (if accurate).

Comments from the reviewers:

Reviewer #1: The authors report a Mendelian randomization study to assess the causal effect of higher BMI on QALYs and total health care costs, and to assess the cost-effectiveness of two interventions that lower BMI, including laparoscopic surgery and restriction of volume promotions for high fat, salt and sugar products, on QALYs and total health care costs. Genetic variants identified in genome-wide association studies for BMI are applied as instrumental variables in analyses of 310,913 participants of the UK Biobank. The authors find that each unit increase in BMI is causally associated with 0.0065 decrease in QALYs per person per year, and 42.23£ increase in total health care costs per person per year. They estimate that laparoscopic surgery over 20 years would lead to a total increase in QALYs of 0.92 per person and a decrease in total healthcare costs of 5,096£ per person. They also estimate the restricting volume promotions for high fat, sat and sugar products across 21.7 million adults in 40-69 years in England and Wales would increase QALYs by 20,551 per year and decrease health care costs by 137£ million per year.

Overall, the paper is very relevant for the readership and carefully written, and demonstrates the relevance of the Mendelian randomization method as a potential new adjunct for decision-making relating to health care policies. While there are multiple limitations and assumptions to the cost-effectiveness estimations and to the Mendelian randomization method itself, I consider that they are comprehensively and appropriately discussed in the paper. Thus, I'm pleased to recommend the paper to be accepted for publication.

Reviewer #2: The present manuscript aims at examining causal relationship between changes in BMI levels and quality of life and healthcare cost through Mendelian randomization (MR) approach using instrument variable regression analyses. The authors have demonstrated how available genetic datasets can be utilized for deriving causal inferences for cost effectiveness of public health interventions. This approach is immensely useful as it requires comparatively less resources and time a compared to other methods like trials and cohorts that are expensive to conduct and are more time consuming. However, the MR findings should be carefully interpreted as they depend on the robustness of the instrument variable (genetic proxy) of the exposure on causal mechanism being examined. The authors have described how they ruled out the bias of pleiotropy as well which is very crucial in MR analyses. The authors have used very robust instrument fr BMI using established GWAS loci while excluding recent studies that included UK Biobank population. Apart from examining the causal effect of causal effect of BMI on QALYs and healthcare costs, they examined the effect on MR results on different policies as well that are immensely informative for policy advocacy. The detailed methodology and the sensitivity analyses demonstrates authors thorough understanding of MR technicalities and their ability to utilize MR approach to answer research questions of public health relevance.

I have no specific queries or recommendations to revise the manuscript. This manuscript can be accepted in its present form.

Reviewer #3: This an exhaustive study, with an extended number of secondary and sensitivity analyses. The Mendelian Randomization analyses with individual level data and with summarized data are correctly applied.

The present work is relevant since it has the potential to inform policies that improve the quality of life and the healthcare costs by taking measures that reduce the BMI of the population.

Generally, I feel that the manuscript should be written in a clearer manner. In addition, I have other comments:

1. The results section is very dense and hard to follow in some parts. Authors might consider to re-write it a bit.

2. I do not understand why the QALYs units are sometimes expressed in % and sometimes in costs per year (pounds). Please clarify.

3. In “Policy Analyses, section d)”, authors say that if all participants with BMI >25kg/m2 had a BMI of 25kg/m2, the QALYs would be decreased. Is it correct like this? One would expect that if all people changed from overweight to normal weight, the quality of life would increase.

4. Table 2 and Figures 2 and 3 have slightly different numbers for the overall results. Please double check.

5. I feel that a discussion of the results from a clinical point of view or comparing the results with previous studies is missing in the results section.

Reviewer #4: This paper describes a study that used Mendelian randomization to estimate the effect of high body mass on (monetized) quality of life and health care costs and estimate the economic merit of a few policy-relevant scenarios. The paper is eloquently written and uses innovative methods to address an issue of large public health relevance. I have a few questions for clarification but found this to be a strong study overall.

The study presents remarkable findings regarding the effect of BMI on outcomes that is not via the usually modelled major disease groups (which is of clear relevance to modellers like me). However, that effect seems related to the imputation of quality of life estimates for each participant, which was done using Sullivan's method, rather than from the Mendelian randomization. Is that correct?

The QALY percentages used to express the impact of changes in BMI on quality of life, are those percentage-points (with 100%, or 1 QALY, as basis), or relative percentages (with 'current'/prior QALY values as basis)?

Were life years lost accounted for in the QALYs? How - by dividing QALYs by 20 even if life was cut short? Was each death valued as 1 QALY lost per year till the end of follow-up (which would underestimate losses) or was it multiplied by life expectancy at the age of death (with what value?)? I would expect the former to be the case, consistent with the 20-year time horizon. Which of course means that while most or all of the costs of the interventions have been taken into account in these analyses, the benefits have been understated, and the true cost-effectiveness of interventions will have been underestimated.

In contrast, the assumptions regarding bariatric surgery, notably the assumption of no adverse side-effects and no peri-operative mortality, will have resulted in some degree of overestimation of the economic credentials of this intervention.

Does the Mendelian randomization approach produce realistic results, since most (if not all) interventions to reduce weight act via changes in diet or physical activity? Large as the results are, if they reflect purely the effect of extra body mass (and not also those of, say, lower sugar consumption or more physical activity) are they not likely underestimates of the true gains that can be expected from interventions?

Minor comments

Typo 'BNI' on page 12 of the supplementary file.

[LINK]

---

## [Decision Letter · Decision Letter 2]

21 Jun 2021

Dear Dr. Harrison,

Thank you very much for re-submitting your manuscript "Long term cost effectiveness of interventions for obesity: a Mendelian randomization study" (PMEDICINE-D-20-02167R2) for review by PLOS Medicine.

I have discussed the paper with my colleagues and the academic editor and it was also seen again by two reviewers. I am pleased to say that provided the remaining editorial and production issues are dealt with we are planning to accept the paper for publication in the journal.

[LINK]

We look forward to receiving the revised manuscript by Jun 28 2021 11:59PM.   

Sincerely,

Caitlin Moyer, Ph.D.

Associate Editor 

PLOS Medicine

plosmedicine.org

Requests from Editors:

1. Data availability statement: Please change “will be” to “is” in the sentence: “The empirical dataset is archived with UK Biobank and made available to individuals who obtain necessary permissions…” and ensure that this is true.

2. Title: Please capitalize the first word of the subtitle: “Long term cost effectiveness of interventions for obesity: A Mendelian randomization study”

3. Abstract: Lines 31-37: Please provide additional clarification or details, as much as possible, throughout the abstract on how MR analysis was used, because the application of Mendelian randomization techniques to evaluate cost effectiveness may be unfamiliar for readers. Please note that PRS for BMI is used as the instrumental variable for the Mendelian randomization analysis.

4. Abstract: Methods and Findings: Line 62: Please slightly clarify the sentence “Large sample sizes are required for sufficient statistical power.” to indicate if this is describing one of the limitations of your study, or general limitation of MR studies.

5. Author summary: In the section "What did the researchers do and find?" please consolidate to 3-4 bullet points for this section, if possible. In the section “What do these findings mean?” we suggest an additional point, summarizing the broad implications of the findings for public health, clinical practice, or policy, relating back to the main research questions.

6. Box 1: We suggest replacing “cheap” and “cheaper” with “inexpensive” or “less expensive” where appropriate.

7. Methods: Section “Data and Code Availability” Please report the information here earlier in the Methods section.

8. Results: Line 358: Please use “was associated with” rather than “caused” as the MR analysis provides evidence in support of causal associations.

9. Discussion: Line 553: We suggest revising to “...our results suggest that obesity may be causally related to inequalities in quality of life.” or similar.

10. Conclusion: We suggest adding a sentence or reorganizing the paragraph to touch on additional conclusions of the study- perhaps by highlighting the statement “The effect of increasing BMI on health related quality of life may be larger than previously thought…” earlier on in the paragraph, or summarizing the health-related implications in addition to the methodological advantages/advance of the MR analysis.

11. Acknowledgements: Please make sure the funding information is included in the “Financial Disclosures” section of the manuscript submission form.

12. References: Please double check that the "Vancouver" style is used for reference formatting, and see our website for other reference guidelines https://journals.plos.org/plosmedicine/s/submission-guidelines#loc-references. Please double check the formatting of: 21, 25, 32. Please check for updated citation information for articles listed as preprints. For reference 39, please provide the updated citation information and if this is not available, please provide an alternate reference. Please note that articles cannot be listed in the reference list until they have been accepted for publication or are publicly available on a preprint archive.

13. Figure 1: Please define all abbreviations used (BMI, QALY, CVD, RCT) in the figure legend.

14. Figure 2 and Figure 3: In the legend, please indicate the adjusted-for variables for the multivariable adjusted analyses. Please define all abbreviations used, such as QALY and BMI in the legend.

15. Figures 4 and 5: Please define the abbreviations “BMI” and “QALY” in the legends.

16. Table 2: Please note in the legend the variables adjusted for in the multivariable adjusted analysis.

17. Supplementary Figure S2 and S3: Please define abbreviations used in the legends, including BMI, QALY, and PRS.

18. STROBE Checklist: Please make it clear, where you are referring to numbers, that these represent paragraphs (for example, Discussion, 1-2 could be Discussion, paragraphs 1-2).

19. Supplementary Tables: Thank you for including the legends for the Supporting Information Tables. We would suggest the titles/legends also be included with each table.

Comments from Reviewers:

Reviewer #3: I do not have further comments for the authors. I believe the manuscript is now acceptable for publication. 

Reviewer #4: You have responded well to my previous comments. Congratulations on this very interesting paper.

[LINK]

---

## [Editor Report · Decision Letter 3]

30 Jun 2021

Dear Dr. Harrison,

Thank you very much for re-submitting your manuscript "Long term cost effectiveness of interventions for obesity: A Mendelian randomization study" (PMEDICINE-D-20-02167R3) for review by PLOS Medicine.

I have discussed the paper with my colleagues and the academic editor, and provided the remaining minor editorial and production issues are dealt with we are planning to accept the paper for publication in the journal.

[LINK]

We look forward to receiving the revised manuscript by Jul 07 2021 11:59PM.   

Sincerely,

Caitlin Moyer, Ph.D.

Associate Editor 

PLOS Medicine

plosmedicine.org

Requests from Editors:

1. Thank you for clarifying the methodology presentation in the abstract. We feel that this is important for understanding the study. However, while the abstract is thoroughly presented, we request that you please revise to shorten the abstract to no more than 500 words. 

-We suggest summarizing the presentation of the results for the simulations of BMI-targeting interventions, presenting fewer details, such as: 

“We estimated that both laparoscopic bariatric surgery among individuals with BMI greater than 35 kg/m2, and restricting volume promotions for high fat, salt, and sugar products would increase QALYs and decrease total healthcare costs.”

-We suggest similarly summarizing or removing the results describing the decrease in QALY and increase in costs estimated for the increases in BMI between 1993 and 2017, and the decreases in QALYs and costs associated with universal BMI below 25 kg/m2: 

 “Between 1993 and 2017 in England and Wales, the increase in BMI of people aged 40 to 69 years led to a decrease of 1.13% of a QALY per person per year (95% CI: 0.90% to 1.38%) and an increase in annual healthcare costs of £69 per person (95% CI: £53 to £84). 

Compared to if all people with a BMI above 25 kg/m2 aged 40 to 69 years in England and Wales in 2017 had a BMI of 25 kg/m2, QALYs are decreased by 580,494 in total per year (95% CI: 457,907 to 717,691) and annual healthcare costs are increased by £3.58 billion (95% CI: £2.75 billion to £4.34 billion).“

-We suggest removing the following sentence from the limitations: “Sample sizes typically must be larger to achieve the same level of statistical power as in corresponding observational studies.” as this seems to be a very general limitation.

2. Methods: Line 338: Please double check the link referring the reader to information on patient/public involvement, the second link does not seem to work.

3. References: Please update references 23 and 54 with the complete information.

[LINK]

---

## [Editor Report · Decision Letter 4]

9 Jul 2021

Dear Dr Harrison, 

On behalf of my colleagues and the Academic Editor, J. Lennert Veerman, I am pleased to inform you that we have agreed to publish your manuscript "Long term cost effectiveness of interventions for obesity: A Mendelian randomization study" (PMEDICINE-D-20-02167R4) in PLOS Medicine.

PRESS

Sincerely, 

Caitlin Moyer, Ph.D. 

Associate Editor 

PLOS Medicine